# Infections Caused by *Moellerella wisconsensis*: A Case Report and a Systematic Review of the Literature

**DOI:** 10.3390/microorganisms10050892

**Published:** 2022-04-24

**Authors:** Daphnie Germanou, Nikolaos Spernovasilis, Anastasios Papadopoulos, Sofia Christodoulou, Aris P. Agouridis

**Affiliations:** 1Department of Internal Medicine, German Oncology Center, Limassol 4108, Cyprus; dafni.germanou@goc.com.cy (D.G.); sofia.christodoulous@goc.com.cy (S.C.); 2Department of Infectious Diseases, German Oncology Center, Limassol 4108, Cyprus; nikspe@hotmail.com; 3School of Medicine, University of Crete, 71003 Heraklion, Greece; 4Department of Medical Oncology, German Oncology Center, Limassol 4108, Cyprus; anastasios.papadopouloss@goc.com.cy; 5School of Medicine, European University Cyprus, Nicosia 2404, Cyprus

**Keywords:** *Moellerella wisconsensis*, emerging pathogen, case report, systematic review

## Abstract

Background: *Moellerella wisconsensis*, a member of the family of *Enterobacteriaceae*, although isolated widely in nature, rarely causes infections in humans. Herein, we report a case of isolation of *M. wisconsensis* from pigtail end culture, urine culture and blood culture in a 76-year-old patient. Objective: To systematically address all the relevant information regarding *M. wisconsensis* through literature. Methods: We searched PubMed and Scopus databases up to January 2022 and performed a qualitative synthesis of published articles reporting infection from *M. wisconsensis* in humans. Results: We identified 25 records on PubMed and 43 additional records on Scopus. After removing duplicates, we examined in detail 15 articles. Ten studies with a total of 17 cases were included in our systematic review. Nine studies described isolated case reports, while 1 study described 8 cases. The origin of the infection was the alimentary tract in 9 cases, gallbladder in 4 cases, peritoneal cavity in 2 cases, respiratory tract in 1 case and hemodialysis catheter insertion site in 1 case. In 3 of the aforementioned cases *M. wisconsensis* was also isolated in blood cultures. Conclusion: Physicians should be aware that *M. wisconsensis* can be present in multiple clinical specimens and that the antibiotic resistance profile of the isolates may pose significant challenges.

## 1. Introduction

*Moellerella wisconsensis* is a member of the family of *Enterobacteriaceae*. The name was proposed by Hickman-Brenner et al. in 1984, after the Danish microbiologist Vagn Møller and, until then, 6 of the 9 known strains had been isolated in Wisconsin, USA [1]. The microorganism has been isolated widely in nature [1,2,3,4,5,6,7,8], but only rarely from human clinical samples.

Herein, we report a case of infection due to *M. wisconsensis* which was isolated from percutaneous nephrostomy pigtail end culture, urine culture and blood culture. In addition, we systematically address through literature searching all the relevant evidence regarding *M. wisconsensis* infections.

### Case Presentation

A 76-year-old man with a history of prostate adenocarcinoma infiltrating the urinary bladder and the rectum, and with lymph node, pulmonary and osseous metastases, complicated with obstructive uropathy, was admitted due to fever (39 °C) and rigors that started two days before hospital admission, while three days before admission, he had undergone percutaneous placement of ureteral double J stents and removal of nephrostomy catheters bilaterally. During the procedure, the pigtail ends were cut off and placed in the culture media for bacterial evaluation.

On admission, a temperature of 37.5 °C was recorded. Urinalysis, urine culture, blood culture and blood tests were drawn. Laboratory tests revealed an abnormal C-reactive protein (248.1 mg/dL) and renal impairment (Urea: 131 mg/dL, Cr: 2.69 mg/dL). Computed tomography of the abdomen-pelvis without contrast showed a dilated urine bladder with absence of hydronephrosis. Ultrasonography revealed a high amount of post void urine in the bladder. The pigtail was removed, and a Foley catheter was placed, leading to drainage of approximately 1000 mL of urine. Pigtail end, urine and blood cultures tested positive for *M. wisconsensis* by BD PhoenixTM-100 Automated Microbiology System using panel NMIC/ID-55 (Gram-negative susceptibility card), further confirmed by 16S rRNA sequencing. The isolate was resistant to several antibiotics, including penicillins, cephalosporins, co-trimoxazole, aminoglycosides, fluoroquinolones and colistin (Appendix A). Based on the phenotypic susceptibility results, which were verified with broth microdilution method (for colistin) and Etest (for piperacillin-tazobactam and carbapenems), targeted treatment with piperacillin/tazobactam was initiated, and the patient recovered uneventfully and was discharged on hospital day 8.

## 2. Materials and Methods

### 2.1. Study Design and Aims

We performed a qualitative synthesis of published articles reporting infection from *M. wisconsensis* in humans. The purpose of this systematic review is to evaluate and better understand the pathogenicity of this rare microorganism. The idea was conceptualized after the finding of *M. wisconsensis* in urine and blood cultures of the above reported patient.

### 2.2. Search Strategy

An extensive bibliographic search of Medline via PubMed and Scopus databases was conducted from inception until 15 January 2022. No language restriction was performed. Initial searches were done using the following search terms: “*Moellerella*” AND/OR “*Moellerella wisconsensis*”. Additional studies were identified from the references provided by retrieved studies.

### 2.3. Eligibility Criteria

Inclusion criteria for our systematic review encompasses articles including at least one case of *M. wisconsensis* infection. Only papers based on humans were considered eligible.

### 2.4. Data Extraction

Studies were independently and thoroughly examined by two investigators (D.G., A.P.A.) and studies’ characteristics were extracted. We evaluated studies’ data (first author, publication year, study design, country), and patients’ characteristics (age, sex, clinical presentation, infection site). Any discrepancy between the reviewers was resolved by consensus. For the review of our analysis, which was designed according to the guidelines of 2020 [9], data extraction was performed with adherence to Preferred Reporting Items for Systematic reviews and Meta-Analysis (PRISMA model). Due to the study design, no Institutional Review Board (IRB) approval was obtained. Of note, for our case report, a patient’s informed consent was retrieved.

### 2.5. Assessment Risk of Bias

A systematic assessment of bias in the included studies was performed using the Joanna Briggs Institute (JBI) critical appraisal checklist for case reports [10]. The items used for the assessment of each study were as follows: patient’s demographic characteristics, patient’s history, patient’s current clinical condition, diagnostic tests or assessment methods and the results, the intervention(s) or treatment procedure(s), post-intervention clinical condition, adverse events (harms) or unanticipated events and takeaway lessons. According to the recommendations of the JBI’s tool for assessing case reports, a judgment of “yes” indicated low risk of bias, while “no” to any of the included questions negatively impacts the overall quality of the case reports. Labelling an item as “unclear” indicated an unclear or unknown risk of bias. Risk-of-bias assessment was performed independently by 2 reviewers (D.G., A.P.A.); disagreements were resolved by consensus.

## 3. Results

### 3.1. Study Selection

In Figure 1, the PRISMA flow chart reveals how the selection of our studies was made. With the above-mentioned search terms, we identified 25 records on Medline via PubMed and 43 additional records on Scopus. After detecting and removing duplicates, we had 43 articles, of which we initially excluded 28 because of reviews and trial design. Subsequently, we examined in detail the remaining 15 articles. Among them, 1 study could not be retrieved, and 4 trials were rejected because selection criteria were not met (Appendix A). Finally, 10 studies of a total 17 case reports (patients with *M. wisconsensis* infection) were included in our systematic review.

### 3.2. Study Characteristics

The included studies were published between 1984 and 2020 (Table 1). Nine studies described isolated case reports, while 1 study described 8 case reports. The latter cases were isolated in the US (6 in Wisconsin, 1 in New York and 1 in Virginia). Regarding isolated case reports, three studies were conducted in France, while the remaining 6 cases were reported in Germany, Belgium, Czech Republic, Spain, Turkey and India (1 each, respectively). Of the ten included studies totally, 7 were written in English, while 2 were written in French and 1 in Turkish language.

### 3.3. Origin of the Infection

The origin of the infection was the alimentary tract in 9 cases, gallbladder in 4 cases, peritoneal cavity in 2 cases, respiratory tract in 1 case and hemodialysis catheter insertion site in 1 case. In three of the aforementioned cases, *M. wisconsensis* was also isolated in blood cultures (Table 1).

### 3.4. Quality Appraisal

The overall quality of the cases was good, as most articles were determined to have low risk of bias, while only one study, which included 8 cases [1], was identified as having a high risk of bias. These results are included in Table 2.

**Table 1 microorganisms-10-00892-t001:** Study (Case reports) Characteristics of *Moellerella wisconsensis* infections reported in the literature.

Author	Year	Study Design	Country	Patient Age/Sex	Clinical Presentation	Sample
Hickman-Brenner [1]	1984	Case Series	Wisconsin, USA	5 y/o ♀	Diarrhea	Feces
Hickman-Brenner [1]	1984	Case Series	Wisconsin, USA	29 y/o ♂	Diarrhea	Feces
Hickman-Brenner [1]	1984	Case Series	Wisconsin, USA	40 y/o ♂	Bloody diarrhea	Feces
Hickman-Brenner [1]	1984	Case Series	Wisconsin, USA	62 y/o ♂	Gastroenteritis	Feces
Hickman-Brenner [1]	1984	Case Series	Wisconsin, USA	16 y/o ♀	Not reported	Feces
Hickman-Brenner [1]	1984	Case Series	Wisconsin, USA	38 y/o ♀	Not reported	Feces
Hickman-Brenner [1]	1984	Case Series	Virginia, USA	NA	Diarrhea	Feces
Hickman-Brenner [1]	1984	Case Series	New York, USA	NA	Not reported	Feces
Wittke [11]	1985	Case Report	Hamburg, Germany	71 y/o ♂	Acute cholecystitis	Bile
Ohanessian [12]	1987	Case Report	France	77 y/o ♀	Acute cholecystitis	Bile
Kubiniek [13]	1995	Case Report	France	67 y/o ♀	Small bowel perforation with peritonitis	Peritoneal fluid
Cardentey-Reyes [14]	2009	Case Report	Belgium	46 y/o ♂	Acute cholecystitis with peritonitis and secondary bacteremia	Bile, blood
Wallet [15]	1994	Case Report	Czech Republic	20 y/o ♀	Inhalation pneumonia, deep coma	Bronchial aspirate
Aller [16]	2009	Case Report	Spain	80 y/o ♂	Acute cholecystitis with secondary bacteremia	Bile, blood
Seyman [17]	2013	Case Report	Turkey	53 y/o ♀	Central venous catheter-related bloodstream	Pus from hemodialysis catheter insertion site, blood
Leroy [18]	2016	Case Report	Nantes, France	64 y/o ♂	Small bowel perforation with peritonitis	Peritoneal fluid
Ahmad [19]	2020	Case Report	India	14 d/o ♀	Diarrhea	Feces
Germanou	2022	Case Report	Limassol, Cyprus	76 y/o ♂	Urinary tract infection with secondary bacteremia	Pigtail end, urine, blood

NA: Not applicable.

**Table 2 microorganisms-10-00892-t002:** Reported cases and their risk of bias according to the Joanna Briggs Institute (JBI) Critical Appraisal Checklist for Case Reports [10].

Author	Year	Were Patient’s Demographic Characteristics Clearly Described?	Was the Patient’s History Clearly Described and Presented as a Timeline?	Was the Current Clinical Condition of the Patient on Presentation Clearly Described?	Were Diagnostic Tests or Assessment Methods and the Results Clearly Described?	Was the Intervention(s) or Treatment Procedure(s) Clearly Described?	Was the Post-Intervention Clinical Condition Clearly Described?	Were Adverse Events (Harms) or Unanticipated Events Identified and Described?	Does the Case Report Provide Takeaway Lessons?	Risk of Bias
Ahmad [19]	2020	yes	no	no	yes	yes	yes	no	yes	Low
Leroy [18]	2016	yes	no	yes	yes	yes	yes	no	yes	Low
Seyman [17]	2013	yes	yes	yes	yes	yes	yes	no	yes	Low
Aller [16]	2009	yes	yes	yes	yes	yes	yes	no	yes	Low
Cardentey-Reyes [14]	2009	yes	yes	yes	yes	yes	yes	no	yes	Low
Kubiniek [13]	1995	yes	yes	yes	yes	yes	yes	no	yes	Low
Wallet [15]	1994	yes	no	yes	yes	no	yes	no	yes	Low
Ohanessian [12]	1987	yes	yes	yes	yes	yes	yes	no	yes	Low
Wittke [11]	1985	yes	no	yes	yes	yes	yes	yes	yes	Low
Hickman-Brenner [1]	1984	yes	no	yes	no	no	no	no	yes	High
		yes	no	no	no	no	no	no	yes	
		yes	no	no	no	no	no	no	yes	
		no	no	no	no	no	no	no	yes	
		yes	no	no	no	no	no	no	yes	
		no	no	no	no	no	no	no	yes	
		no	no	no	no	no	no	no	yes	
		no	no	no	no	no	no	no	yes	

Hickman-Brenner study [1] included 8 cases.

## 4. Discussion

This systematic review focuses on infections by *M. wisconsensis* in humans. To the best of our knowledge, this is the first systematic review conducted on this rare microorganism.

*M. wisconsensis* identification is somewhat difficult, and strains of this microorganism may have been misidentified on several occasions as *Escherichia coli* or *Klebsiella pneumoniae* subsp. *ozaenae* [20]. Specific features of the bacterium include the following: negative for indole production, Voges-Proskauer, H2S production, urea hydrolysis, phenylalanine deaminase, lysine and ornithine decarboxylases, arginine dihydrolase, gas production from D-glucose, acid production from trehalose and motility; positive for the utilization of citrate (Simmons) and acid production from lactose and raffinose [11].

Although it is an uncommon bacterium in the daily clinical practice, there are several case reports in the literature describing its isolation from clinical specimens, such as human stool, bile, blood, bronchial aspirate, and wound swab, and its relationship with clinically overt disease [1,13,14,15,16,17,18,19,21]. Regarding our report, it represents the fourth case of isolation of *M. wisconsensis* from blood culture and the first case of its isolation from pigtail end and urine cultures. The strain was susceptible to β-lactam/β-lactamase inhibitor combinations, aztreonam, carbapenems and tigecycline. On the other hand, resistance was observed to aminoglycosides, fluoroquinolones, co-trimoxazole and colistin. After the administration of piperacillin/tazobactam, the patient recovered promptly.

Regarding the antibiotic susceptibility profile of this microorganism, it shares many common features with other *Enterobacteriaceae*. Specifically, it is naturally susceptible to tetracyclines, aminoglycosides, β-lactams (except oxacillin and benzylpenicillin) fluoroquinolones, chloramphenicol, folate-pathway inhibitors, and nitrofurantoin. On the contrary, it is naturally resistant to oxacillin and benzylpenicillin, macrolides, streptogramins, lincomycin, rifampicin, fusidic acid, glycopeptides, oxazolidinones and colistin [1,20].

However, acquired resistance of this bacterium against several classes of antibiotics may emerge, and multi-drug or extended-drug resistant strains have been isolated in many cases, including the case reported here. The most worrisome of all is that there has been already described cases of infections with *M. wisconsensis* strains that harbored plasmids containing genes that conferred resistance to carbapenems, such as blaNDM-1 and blaVIM-1 [19], and these plasmids could be transferred easily between bacteria within this species. These genes confer resistance to almost all β-lactams, which is extremely alarming considering the intrinsic resistance of this bacterium to colistin. All these taken together reduce the available treatment options for patients suffering infections by multi- or extended-drug resistant strains of *M. wisconsensis*, while requiring prompt implementation of enhanced infection control measures in the health-care environments.

Lack of a greater number of cases and series is a potential limitation in our systematic review. The small number of existing case reports could be attributed to publication bias. In addition, since our final selection was limited to cases and series, it was not possible to carry out a meta-analysis. Furthermore, our protocol was not registered on PROSPERO. The above limitations could result in reaching less robust conclusions. However, by using the JBI critical appraisal checklist for each case report we included in our systematic review, we assessed the methodological quality of each study and managed to improve our systematic review quality.

## 5. Conclusions

In conclusion, physicians should be aware that *M. wisconsensis* can be present in multiple clinical specimens, including urine and blood, causing life-threatening infections if not adequately treated. Possible acquired resistance to carbapenems, along with the intrinsic resistance to colistin, can make the management of these infections challenging. Health-care providers should have all these in mind when treating patients with infections caused by this uncommon microorganism, in order to achieve a favorable outcome for their patients.

## Figures and Tables

**Figure 1 microorganisms-10-00892-f001:**
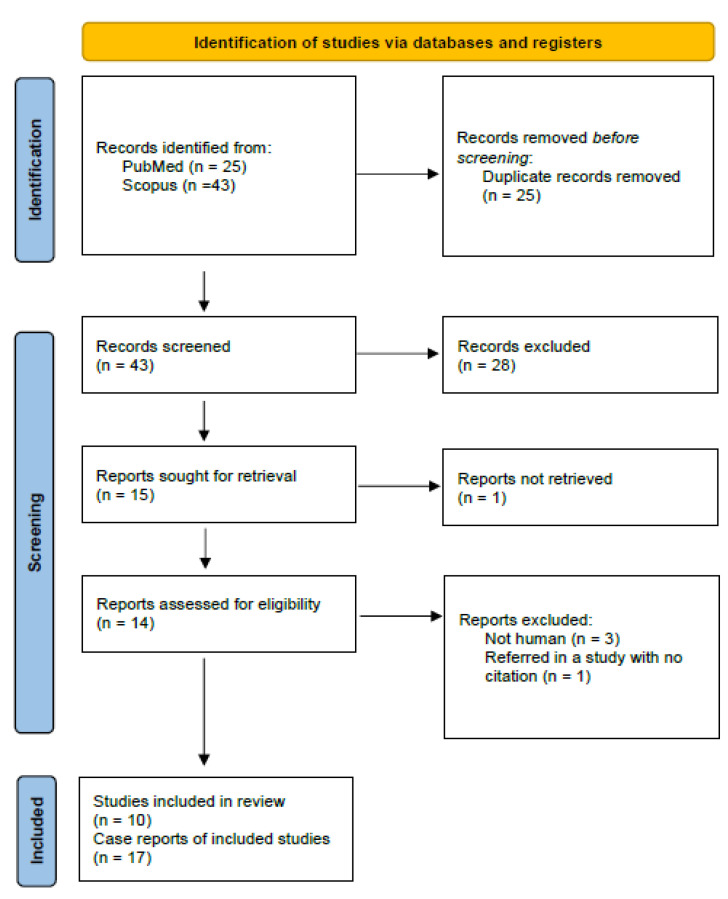
PRISMA flow diagram of articles related to *Moellerella wisconsensis* case reports.

## Data Availability

Not applicable.

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
