# Peer review of "Infections Caused by Moellerella wisconsensis: A Case Report and a Systematic Review of the Literature"

_microorganisms, 2022, doi:10.3390/microorganisms10050892_

Round 1
Reviewer 1 Report
please rename the paragaph 2.1 in study design and aims
did the authors register the protocol of this review? if yes please add information, if not please specify
please detail the inclusion/exclusion criteria in terms of PICOS as suggested by PRISMA guidelines.
line 74 "studies’ characteristics were extracted": could you detail which data were extracted?
Could you explain why Hickman-Brenner study has more than one row in Table 2?
Please add a table with summary descriptive main characteristics of included studies.
Please also add a table with reasons of exclusion.
Please, follow the PRISMA guidelines more precisely.
lines 164-166 could you explain how did you minimized the risk of bias? Could you comment included studies in light of the Risk of bias results?

Reviewer 2 Report
The manuscript by Germanou et al describes a new case of Moellerella wisconsensis infection in a human and provides a systematic review of the literature. The authors are careful to adhere to the guidelines for such a systematic review and the manuscript is well written. I have a handful of suggestions for language revision.
- L15 perhaps use “information” instead of “evidence” or indicate evidence of what
- L19 Ten studies with a total of 17 cases
- L36 through literature searching
- L41 rigors which started two days earlier. Two days later
- L45 Was the temp 37.5 on admit or 39 (as reported later)
- L46 define CRP
- L86 events and takeaway lessons
- L52-L55 How was sensitivity determined? It would be useful to include MICs in a supplementary Table
- L94 In Figure L100 of a total of 17
- L108 while the remaining 6 cases L109 India (1 each, respectively).
- Table 1 title replace “isolates” with “infections” as the table does not describe the bacterium
- Depending on journal style, I would add spaces after the countries and before the references in Table 1.
- Table 1 Define NA as a footnote
- L124 on infections L128 on several L130 and Voges L130 subscript 2 in H2S
- L141 to aminoglycosides L151 including the case reported here.
- L161 The small number of existing

Round 2
Reviewer 1 Report
Dear Authors, thank you for your efforts in meeting my suggestions. However, I disagree with you referring to table 1. Actually, it does not explain the characteristics of the included studies. On the contrary, it describes the characteristics of cases. Table 1 should contains info on first author, publication year, study design, country, and patients’
characteristics (age, sex, clinical presentation, infection site). Please add
Could you explain in the text (and as a footnote in the table) why Hickman-Brenner study has more than one row in Table 2? thanks
